# Development of the Roe Deer–*Fascioloides magna* Association over Time

**DOI:** 10.3390/pathogens14060516

**Published:** 2025-05-22

**Authors:** Anja France Noëlle Renée Buet, Miljenko Bujanić, Krešimir Krapinec, Ivica Bošković, Anđelko Gašpar, Dean Konjević

**Affiliations:** 1MAJCAN Veterinary Clinic, K. Frankopana 38, 43000 Bjelovar, Croatia; anja.buet@outlook.com; 2Faculty of Veterinary Medicine, University of Zagreb, Heinzelova 55, 10000 Zagreb, Croatia; mbujanic@vef.unizg.hr; 3Faculty of Forestry and Wood Technology, University of Zagreb, Svetošimunska 23, 10000 Zagreb, Croatia; kkrapinec@sumfak.unizg.hr; 4Faculty of Agrobiotechnical Sciences, Josip Juraj Strossmayer University of Osijek, V. Preloga 1, 31000 Osijek, Croatia; ivica.boskovic@fazos.hr; 5Croatian Veterinary Chamber, Heinzelova 55, 10000 Zagreb, Croatia; hvk@hvk.hr

**Keywords:** roe deer, *Fascioloides magna*, host–parasite association, co-evolution

## Abstract

The trematode *Fascioloides magna* is originally a parasite of North American deer species. Upon its arrival to Europe, *F. magna* met new intermediate and final hosts. Depending on the type of host, the clinical picture, pathological findings, epidemiology and outcome can vary significantly. As an aberrant host, it was long believed that the roe deer (*Capreolus capreolus*) fails to develop pseudocysts, and therefore the infected animal dies before the parasite can mature and start to produce eggs. In this study, 676 roe deer livers were collected in Croatia during the hunting years of four consecutive years (2019–2023) in Bjelovar-Bilogora County (BB), and 184 livers were collected from Zagreb County (ZG) in the hunting year 2022/2023. Livers were analysed macroscopically and on a cut surface for lesions and any developmental stage of *F. magna* according to a standard protocol. The mean prevalence of infected livers during the whole study period was 12.86% in BB and 3.8% in ZG. No pseudocysts were detected in samples from ZG, while there was an increasing trend of pseudocyst presence over time in BB. The occurrence of pseudocysts in infected livers showed a rapid increase after the hunting season 2019/2020, before becoming constant (at approx. 40%). The odds of finding pseudocysts ranged between 2.7 (OR = 2.7317, CI 95% 0.3108 to 24.0095, *p* = 0.365) and 2.9 (OR = 2.9167, CI 95% 0.3163 to 26.8924, *p* = 0.345) times higher in later years compared to 2019/2020. Similarly, an increasing trend (though less pronounced) was observed in the numbers of livers simultaneously containing pseudocysts and fluke migratory stages. The results indicate a potential change in the roe deer–*F. magna* association, where an increasing number of roe deer are forming pseudocysts and can survive even multiple infections.

## 1. Introduction

Parasites are ubiquitous organisms, known also as potential agents of disease and even death. However, in addition to this, they are increasingly recognised as integral components of ecosystems and major drivers of biological evolution. By definition, a parasite is an organism that depends on at least one host species for habitat and nutrition during at least one stage of its life [1], potentially decreasing or optimising the fitness of the host. Parasites play several roles in ecosystems. They are the key determinants of animal population dynamics and community structure, forming a sub-class of predation. Parasitism can indirectly affect animal immunity, nutrition, genetics, and behaviour. Though host–parasite interactions are dynamic and frequently rapidly evolving associations [2], the introduction of new parasites to naive host populations can cause detrimental effects and even a local decline of the species. *Fascioloides magna* is known to have a detrimental effect on aberrant hosts considering its size, permanent migration through the liver, and occasional damage to other internal organs [3,4]. This is in accordance with Anderson (1972), who stated that parasites of moderate to low pathogenicity can significantly impact population growth, while highly pathogenic parasites could cause a local extinction of more sensitive host species [5]. This, in turn, can lead also to the extinction of not just hosts, but of radical parasite forms. Despite its potential to harm hosts, it is not in the parasite’s interest to kill its final host, but to survive and complete its life cycle. Therefore, it is understandable that host–parasite interactions are dynamic model systems used in a wide range of ecological and evolutionary studies.

Genetic studies have shown that the large American liver fluke (*Fascioloides magna*) was imported to Europe with American cervids on at least two occasions, resulting in the formation of three permanent foci of infection [6]. Based on the infection characteristics, three types of final hosts are recognised among the European wildlife: typical (red deer and fallow deer), aberrant (roe deer and mouflon), and dead-end (wild boar) hosts [7]. For a time, it was clear that infection in roe deer is fatal due to the inability of the fluke to stop its migration, and consequently a failure of the organism to form the pseudocyst [7]. The prevalence of *F. magna* infection in roe deer is usually lower than in red deer, and can range from 11.11% to 46.1% [8,9,10]. Recent studies have reported potential signs of adaptations of roe deer to *F. magna* (and vice versa) from the viewpoint of pseudocyst formation, the occurrence of chronic infections and even the shedding of the eggs via faeces [11,12,13].

The hypothesis of this study is that with the prolonged presence of this alien parasite in certain geographical areas, both roe deer and *F. magna* will develop mechanisms to survive the infection, and to continue its life, or in case of the parasite, to complete the life cycle.

## 2. Material and Methods

### 2.1. Animals and Locations

A total of 860 roe deer livers were collected in Croatia during the hunting seasons of four consecutive years (2019–2023), of which 676 livers were collected in Bjelovar-Bilogora County (BB), and 184 in Zagreb County (ZG; only in 2022/2023) (Figure 1).

The asterisk in Figure 1 shows the approximate location of the first confirmation of *F. magna* in cervids in Croatia, at the Šeprešhat locality in the Baranja region [14]. Two main rivers, the Danube and Sava River, are presented as solid blue lines, indicating potential main routes of spread of *F. magna* northwards from its entry point into Croatia. Livers were collected during the regular execution of approved Game Management Plans (GMPs) and analysed within the Croatian Science Foundation grant (no. IP-8963). Since sample size and the selection of animals depended on hunters, all available livers were collected (non-probability convenience sampling).

The GMP prescribes the quotas for how many animals of each age or sex can be hunted. Livers were collected immediately upon evisceration, stored in the plastic bags and frozen following each respective hunt. All livers from one hunting season were collected and analysed on the same day. Both BB and ZG are located in central Croatia, and characterised by mainly lowland to hilly relief, rich in springs and watercourses. The natural vegetation is made up of evenly aged forests alternating with grasslands (mostly meadows and arables). This study was approved by the Ethics Committee of the Faculty of Veterinary Medicine, University of Zagreb (Class: 640-01/24-02/08, No.: 251-61-01/139-24-19) on 19 September 2018.

### 2.2. Liver Analysis

Livers originated from adult animals on an anonymous base (no data on sex or exact age were provided with livers). Each liver was analysed macroscopically, and the presence of fibrin deposits, surface irregularities, traces of iron-porphyrin pigment, and loss of translucency of the Glisson’s capsule were noted (Figure 2 and Figure 3).

Following external analysis, each liver was sliced into approximately 2 cm slices, and analysed on both sides. The presence of juvenile or adult flukes, fluke migratory paths, and pseudocysts were noted. Livers containing flukes were considered to be infected. Livers containing both pseudocysts and fluke migratory stages were considered chronically infected.

### 2.3. Data Analysis

The normality of distribution was tested using the Shapiro–Wilk test. Pearson’s correlation coefficient was used to test for a correlation between hunting season (year) and the number of pseudocysts. The comparison of the number of pseudocysts between years was performed with the Kruskal–Wallis test. All analyses were performed using the Statistica package (version 14.0.0.15; TIBCO, Palo Alto, CA, USA, 2020). Descriptive data analysis and linear regression were performed in Excel. The odds ratio was calculated in the free online MedCalc program. The map was developed with the ArcGIS 10.1 program (ESRI, GDi d.o.o., Zagreb, Croatia).

## 3. Results

Results of parasitological analysis are presented in Table 1.

In ZG, the prevalence of infected livers was 3.8%. No pseudocysts were detected in the infected livers, and all positive samples contained only fluke migratory stages. In the BB sample, the prevalence of infected livers increased from 2019/2020 (7.29%) to 2020/2021 (14.9%), after which it remained relatively constant (13.51% in 2021/2022 and 13.31% in 2022/2023). Due to variations in sample size, and the high number of negative livers, pseudocyst presence was calculated as a fraction, with the number of pseudocysts as the numerator, and the number of infected livers as the denominator, and expressed as a percentage. Accordingly, the occurrence of pseudocysts was lowest in 2019/2020 (14.28%) and highest in 2020/2021 (41.66%), before becoming relatively constant (40.0% in 2021/2022 and 39.02% in 2022/2023). The number of pseudocysts by year was normally distributed (W = 0.985; *p* = 0.931. Despite the fact that the regression coefficient was 0.6961, the number of pseudocysts in the liver was not associated statistically with the hunting year (*p* = 0.203). The average number of pseudocysts per liver by hunting year ranged from zero (2019/2020) to 0.423 (2022/2023). Despite this, no statistically significant difference was found between hunting years. An increase in the numbers of fluke migratory stages was expected due to the larger sample size and increased number of chronically infected animals. The highest numbers of migratory stages were detected in 2020/2021 (10.5% of all livers, or 70.83% of infected livers). The lowest prevalence was detected in the first study year (2%; 28.57%). The remaining two hunting years had approximately the same prevalence of migratory stages (5.4% and 6.4% of all livers; 40% and 48% of infected livers, respectively). The presence of migratory stages is highly dependent on the presence of red deer, as the main spreader of *F. magna* eggs in the environment.

Figure 2 shows a typical roe deer liver infected with *F. magna*, without the formation of the pseudocyst, while Figure 3 shows an infected roe deer liver with the formation of the pseudocyst and retained function. In the liver presented in Figure 3, irregularities of the surface are less pronounced. Traces of black pigment called iron-porphyrin are present and are considered a pathognomonic sign of *F. magna* infection. The liver edges are still sharp, and liver size is near normal. Part of the diaphragm is attached to the liver. The cut surface of the liver shows a well-developed pseudocyst. The pale walls of the pseudocyst are characteristic for a longer duration of infection.

The number of infected livers, presented as a proportion with respect to sample size, is presented in Figure 4.

The number of infected roe deer livers almost doubled in 2020/2021, before becoming more constant over the rest of the study period. Figure 5 shows the detected numbers of fluke developmental stages and pseudocysts by hunting year. In general, there was a slight increase in the simultaneous findings of pseudocysts and migratory stages by year.

Over time, the number of infected roe deer livers and pseudocysts increased. This is consistent with the increasing sample size, but could also be related to a prolonged survival of infected animals, thereby increasing the likelihood of being hunted and analysed. Since the formation of pseudocysts is a prerequisite for parasite and host survival, the larger distance between blue and red lines in the last study year (Figure 6) supports this hypothesis. Increasing trend in pseudocyst formation over the years is depicted in Figure 7.

Due to variations in sample size between years, and to get a better insight into the presence of pseudocysts over time, the number of pseudocysts was compared with the number of infected livers (Figure 8).

The prevalence of pseudocysts in roe deer livers collected at BB showed a rapid increase after 2019/2020, before becoming relatively constant (approx. 40%). Another important finding is the simultaneous presence of pseudocysts and fluke migratory stages in the liver. This finding indicates the potential for roe deer to survive multiple infections with *F. magna*.

The odds ratio analysis indicated an apparent increase in the likelihood of pseudocyst presence over time; however, none of the results reached statistical significance. Specifically, the odds of pseudocyst detection were 2.9 times higher in 2020/21 compared to 2019/2020 (OR = 2.9167, CI 95% 0.3163 to 26.8924, *p* = 0.345). In 2021/2020, the odds were 2.8 times higher (OR = 2.800, CI 95% 0.2809 to 27.9081, *p* = 0.380), and in 2022/2023, they were 2.7 times higher (OR = 2.7317, CI 95% 0.3108 to 24.0095, *p* = 0.365) compared to 2019/2020. Therefore, despite the fact that the odds ratio may be biologically suggestive, the lack of statistical significance and wide CI suggest that potential conclusions about host adaptability should be interpreted with caution.

## 4. Discussion

Being originally a parasite of North American deer species, it has been hypothesised that *Fascioloides magna* co-evolved with wapiti (*Cervus elaphus canadensis*) and white-tailed deer (*Odocoileus virginianus*), forming stable host–parasite associations. These hosts can withstand heavy infections and usually show no clinical symptoms [15,16]. Upon its arrival to Europe, the fluke met new, naive populations of intermediate and final hosts. Here, the co-evolution with North American host species became visible through the fact that *F. magna* prefers the mud snail (*Lymnaea truncatula*) and red deer (*Cervus elaphus*) as the main intermediate and final hosts in its life cycle, likely due to the phylogenetic similarity of these species with their North American relatives [17,18]. Other water snails, such as *Lymnaea* spp. and *Radix peregra* can also serve as intermediate hosts, but appear to be less suitable for *F. magna* [19,20,21]. This is in accordance with the statement of Rondelaud et al. (2007) that the giant liver fluke in Europe will likely adapt over time to other snails in the family Lymnaeidae [22].

The final wild hosts in Europe are generally categorised as definitive (red and fallow deer), aberrant (European mouflon, roe deer, experimentally chamois and accidentally Alpine ibex) and dead-end hosts (wild boar) [3,7,16,23], though this terminology varies in the different literature. Clinical signs and pathological findings vary significantly with the type of the host [7]. Definitive hosts usually tolerate infection well, exhibiting visible clinical signs only rarely. Dead-end hosts exhibit no signs and are able even to kill flukes via the formation of thick-walled pseudocysts, while aberrant hosts are the most sensitive, and an infection with a single fluke can be lethal (for a review, see [7]). In aberrant hosts, death occurs mainly as a result of the severe destruction of liver parenchyma due to the permanent migration of the juvenile fluke [24]. Occasionally, death may result also from fluke migration to other organs, usually the lungs or spleen, and consequent infections [4]. However, as with intermediate hosts, it is reasonable to expect that the fluke will adapt to the final host, and vice versa.

Demiaszkiewicz et al. (2018) were among the first to describe the formation of pseudocysts in roe deer [11]. They described five pseudocysts in one roe deer (of 20 analysed) during May 2016 in the Bory Zielonogórskie forest in Poland. Three years later, Konjević et al. (2021) described a chronic *F. magna* infection in roe deer from Croatia for the first time [12]. They analysed 227 roe deer livers from three regions and found 34 positive samples (14.97%). Among positive animals, 14 had pseudocysts (41.17% of positive animals). An interesting finding was the simultaneous presence of the fluke migratory paths, juvenile flukes, and pseudocysts with sexually mature flukes in seven animals. This confirmed the ability of roe deer to survive initial infection, form a pseudocyst and even become infected with new juvenile flukes. However, none of these studies have confirmed *F. magna* eggs in roe deer faeces. This shows that only four years after a comprehensive review on *F. magna* in Europe [7], a significant change in the roe deer–*F. magna* interaction was observed. In the present study, the mean total prevalence of infected roe deer livers in BB was 12.86%, which is similar to the previous report [12]. The lowest prevalence was detected in 2019/2020 (*p* = 7.29%). More positive animals were confirmed in later years, when the prevalence ranged between 13 and 15%. A possible explanation is that roe deer were able to survive *F. magna* infection, allowing for the collection of more positive animals. Contrary to this, the classic course of the disease in aberrant hosts is rapid and carcasses are rarely found, and as such, hunted animals are usually not infected. The prolonged survival of roe deer is confirmed by the increasing trend in the number of pseudocysts during the study period. This increasing trend in pseudocyst formation is also confirmed by the odds ratio that ranged between 2.7 and 2.9-times higher odds that infected animals will have pseudocysts in later years of observation compared to 2019/2020. Observed changes are not statistically significant but are biologically suggestive of potential adaptation processes. Another important indicator of potential changes in the host–parasite interaction is the simultaneous presence of fluke migratory stages and pseudocysts in the liver, indicating that roe deer can form pseudocysts and survive infection, but that they can also function normally and even become re-infected. Figure 1 shows that BB is closer to the origin of fascioloidosis than ZG, suggesting that *F. magna* has likely been present for a longer time in BB, enabling a longer development of this host–parasite association. The larger red deer population in BB also likely contributed to the occurrence and presence of infection. This is further seen by the relatively low prevalence (3.8%) of infected roe deer in ZG in 2022/2023, and that no pseudocysts were found in this region at any time during the study.

The macroscopic analysis of the roe deer livers shows that animals failing to develop pseudocysts succumb to the severe destruction of the liver parenchyma and excessive haemorrhaging. These livers and the parenchyma are damaged by the permanent migration of relatively large parasites. In cases where pseudocysts were formed, livers retained nearly normal shape and function, prolonging the survival of the infected animal. The rapid course of the disease in cases without pseudocyst formation is likely the reason why the main gross lesions on the liver are the destruction of the parenchyma and severe haemorrhaging, rather than signs of an immune reaction. Increased findings of livers with pseudocysts in later years of the study clearly indicate a shift in the host–parasite association.

The adaptation process between roe deer and *Fascioloides magna* is clearly an ongoing process, since recent studies have indicated that *F. magna* can produce eggs in infected roe deer, and that roe deer can occasionally shed these eggs into the environment [13,25]. Interesting observations on this host–parasite association were reported recently by Csivincsik et al. [26]. They stated that the observed changes in the roe deer–*F. magna* association are probably a result of parasite adaptation. In any case, if a pseudocyst is to be formed, it is the fluke that must stop its migration and allow the organism to create this form of barrier. This could also be related to the degree of the host’s immune reaction. Future studies should shed more light on this topic. Finally, this study shows an increasing number of roe deer with pseudocysts were found in the infected areas over time, suggesting a further development of the host–parasite association.

## Figures and Tables

**Figure 1 pathogens-14-00516-f001:**
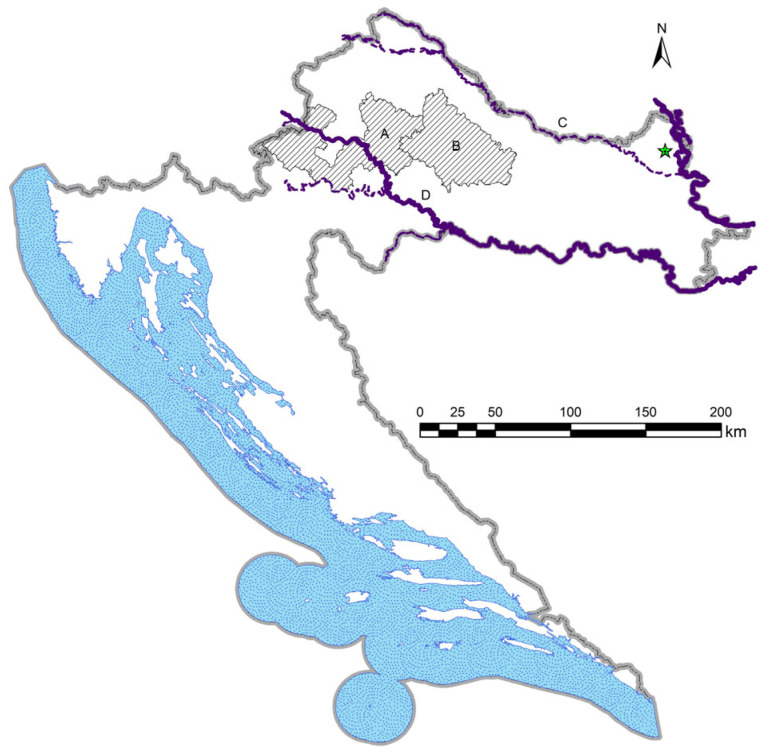
Sampling areas are marked with A (Zagreb County) and B (Bjelovar-Bilogora County). Letters C (Danube River) and D (Sava River) mark the main rivers in this part of Croatia. The asterisk indicates the approximate location of the first record of *Fascioloides magna* in Croatia.

**Figure 2 pathogens-14-00516-f002:**
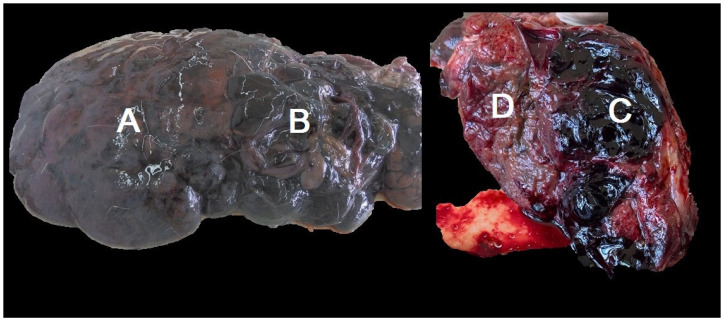
Macroscopic appearance of an infected roe deer liver, and cut surface of the same liver. Note irregularities of the surface and the loss of translucency of the Glisson’s capsule (A), the region of severe tissue destruction and haemorrhage on the intact (B) and cut surface of the liver (C) and the preserved liver parenchyma (D).

**Figure 3 pathogens-14-00516-f003:**
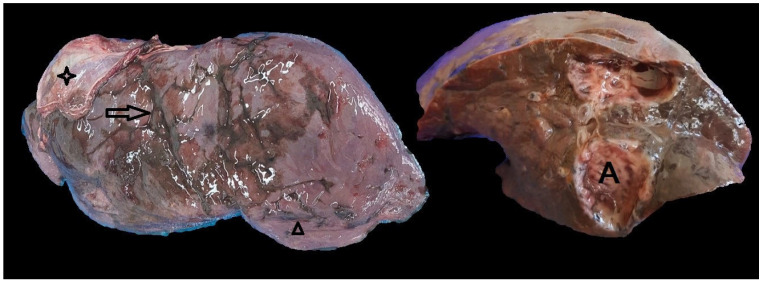
Macroscopic appearance of an infected roe deer liver. On the left image, note iron-porphyrin (arrowhead), surface irregularities (arrow) and remnants of the diaphragm attached to the surface of the liver (four-point star). In the cross-section of the same liver, note the well-developed pseudocyst (A).

**Figure 4 pathogens-14-00516-f004:**
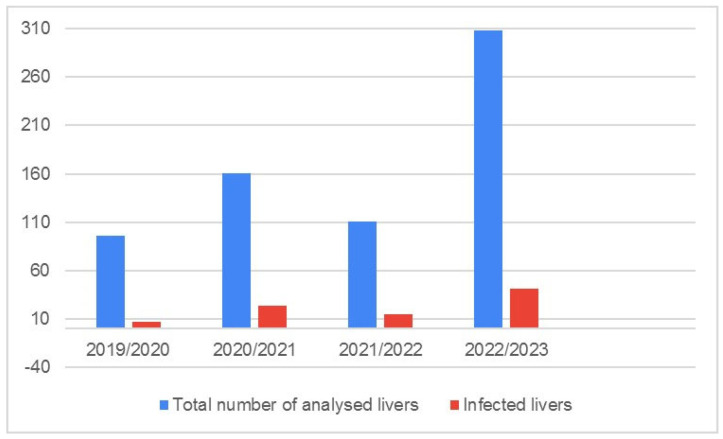
Total number of collected roe deer livers (blue columns) and prevalence of positive livers (red columns), with trendline, from Bjelovar-Bilogora County.

**Figure 5 pathogens-14-00516-f005:**
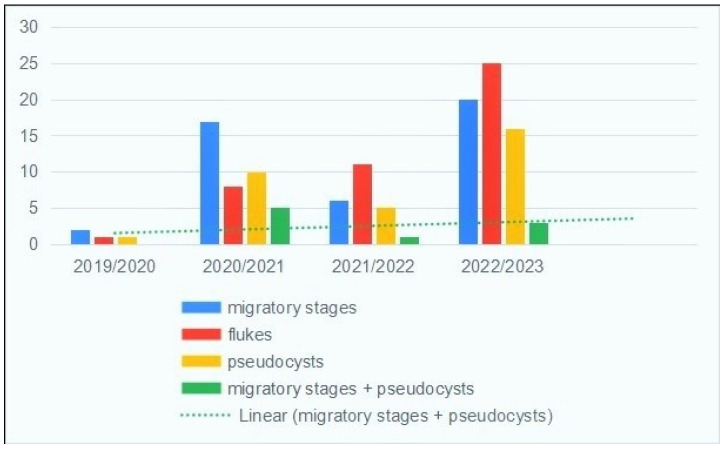
Presence of migratory stages, adult flukes, pseudocysts and livers with both migratory stages and pseudocysts in roe deer collected at BB, during the four study years.

**Figure 6 pathogens-14-00516-f006:**
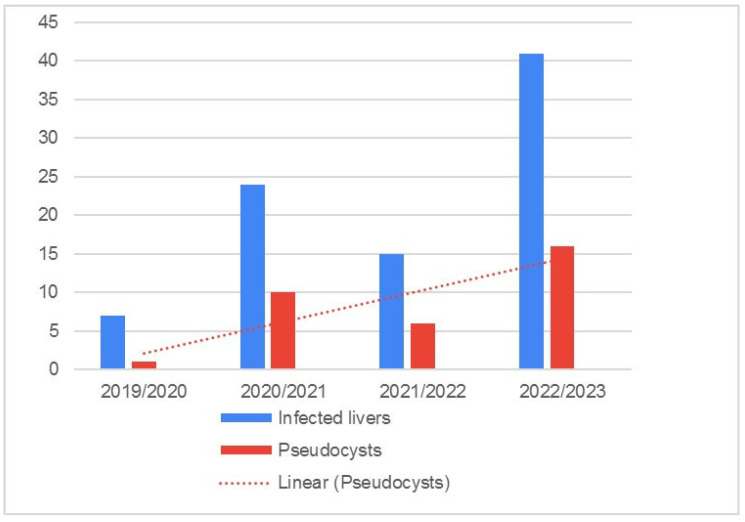
The red line represents the number of pseudocysts found in roe deer livers during the four study years, with trendline; the blue line represents the number of positive livers.

**Figure 7 pathogens-14-00516-f007:**
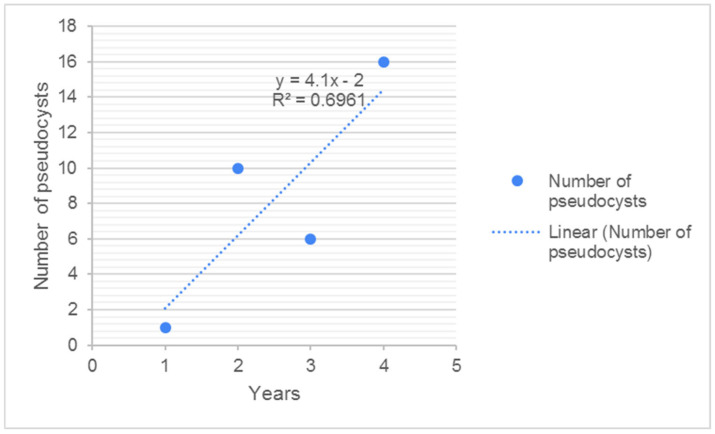
The number of pseudocysts in roe deer livers during the study period from BB, linear regression.

**Figure 8 pathogens-14-00516-f008:**
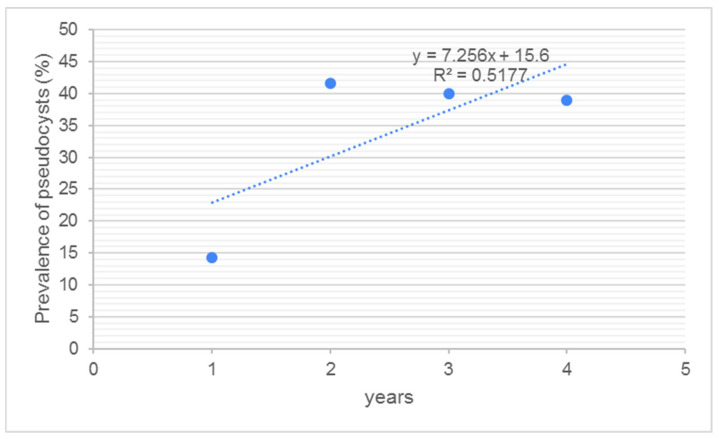
The prevalence of pseudocysts in roe deer livers by hunting year, compared to the number of infected livers.

**Table 1 pathogens-14-00516-t001:** Results of the parasitological analysis of roe deer livers.

Year	Location	Analysed Livers (N)	Infected Livers (N)	Prevalence (%)	Pseudocysts (N)	Prevalence of Pseudocysts with Respect to Positive Livers (%)	Migratory Stages
2019/2020	BB	96	7	7.29	1	14.28	2
2020/2021	BB	161	24	14.9	10	41.66	17
2021/2022	BB	111	15	13.51	6	40	6
2022/2023	BB	308	41	13.31	16	39.02	20
2022/2023	ZG	184	7	3.8	0	0	7

## Data Availability

All data generated or analysed during this study are included in this published article.

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
