# Peer review of "Development of the Roe Deer–Fascioloides magna Association over Time"

_pathogens, 2025, doi:10.3390/pathogens14060516_

Round 1
Reviewer 1 Report
Comments and Suggestions for Authors
Comments and Suggestions for Authors are added in the PDF document.

Author Response
Comment 1: Presented article “Development of the roe deer - Fascioloides magna association over time” (pathogens-3574962) evaluates the gradual changes in the development of roe deer (Capreolus capreolus), one of hosts of trematode giant liver fluke Fascioloides magna, over time. Parasite's findings from two localities in Croatia (Bjelovar-Bilogora County and Zagreb County) were compared over a period of four years. The work is clear, well-structured and the formulations are understandable. The topic of the study is current and of interest of local researchers and veterinarians, as well as for wider scientific community.
Response 1: We thank you for kind words and all suggestions made for improving this article. Also, thank you for observing errors that somehow have occurred in the manuscript.
Comment 2: L. 15 and e.g. 67: Add Latin names of roe deer – Capreolus capreolus in the abstract and in the main text.
Response 2: Done
Comment 3: L. 19, 69: Add a name of locality – Croatia in the abstract and in the Material and Methods.
Response 3: Done
Comment 4: Recommendation for Introduction section (e.g., L. 60): Add information about experimental studies or native observations of the F. magna successful development in roe deer (if available) and indicate the expected changes that could lead to the development of a mechanism of this aberrant host to survive the parasite's infection (in short form to introduce the topic, more discus in Discussion section). Moreover, add literature records and sources, where was roe deer found to be infected until now (e.g. examples of prevalences detected, status of infected animals, clinical signs etc.).
Response 4: Short section is added to the Introduction part. More details are available from cited references. – '' The prevalence of F. magna infection in roe deer is usually lower than in red deer, and can range from 11.11% to 46.1% (8, 9, 10). Recent studies have reported potential signs of adaptations of roe deer to F. magna (and vice versa) in a view of pseudocyst for-mation, occurrence of chronic infections and even shedding of the eggs via faeces (11, 12, 13).
Comment 5: L. 69, 118 (Table 1), 122-136, 156, 197, 201-205, 248, 256, 264: Use the year in full form in the text and tables (as you used for the first time in L. 68 and Figs 4-6), e.g. 23/2023 etc.
Response 5: Done
Comment 6: L. 69, 92, 96, 190: Change the title of the figures to “Figure”, not abbreviated Fig.
Response 6: Done
Comment 7: Figure 1 (L. 79): Although the map was made in ArcGIS, its quality is low and the marked parts are not clearly visible. Please, increase the quality of the figure and mark the Danube and Sava Rivers on it (write names). Increase the size of markings for the sampling areas (A - Zagreb County, B - Bjelovar-Bilogora County) and asterisks.
Response 7: Done
Comment 8: L. 85: Explain, why did you froze the liver, when you analyse them on the same day?
Response 8: Livers were collected during the hunting season and frozen by hunters untill end of the hunting season. We have collected them and analysed. For clarification we added following sentence: '' All livers from one hunting season were collected and analysed on the same day.''
Comment 9: L. 105: Add “were considered to be positive”.
Response 9: Done
Comment 10: L. 117 or Table 1 (L. 118): Add information about studied roe deer, e.g. gender (male/female), age, number of individuals examined (this number may not correlate with the number of analysed livers), part or whole liver etc.
Response 10: Livers were collected on anonimous base from adult animals. Sentence was added: '' Livers originated from adult animals on anonymous base (no data on sex or exact age were provided with livers).''
Comment 11: In Table 1, there is an error in the recalculation of the "Prevalence of pseudocysts with respect to positive livers" in years 2020/2021 (43.47%), correct it to 41.6% (and also then in the text L. 128).
Response 11: Corrected. Thank you for observing this.
Comment 12: L. 129-131: Put the results of the statistical analysis in form of table for better clarity with a reference to the table in the text. Similarly, L. 201-206, the results should be in tabular form.
Response 12: Data are left in the form of text to avoid too many tables and figures/graphs in the manuscript. Section L 201-206 is rephrased.
Comment 13: Figure 4 (L. 152): Change the graph, or divide it into two parts, because "number" is given in whole units, while "prevalence" in percentages.
Response 13: Corrected
Comment 14: L. 176: Remove the title of the graph, because it is not necessary. The legend with the same description is to the right of the graph.
Response 14: Done
Comment 15: L. 173-185: If the number of pseudocysts is shown on the graph (Figure 7), it should correlate with the data in Table 1. The number of pseudocysts in the second and third year of analysis (in BB) should be 10 and 6, respectively, which is not shown correctly in Figure 7.
Response 15: Corrected. Thank you for noticing this.
Comment 16: L. 190-192: The X-axis title should present "year", not "season" (as in Table 1) and should be displayed in whole units (as in Figure 7).
Respoonse 16: Done
Comment 17: L. 280: Reformulate the sentence in the correct English form.
Response 17: Done
Comment 18: There are many formal mistakes in References section, need to be repaired
Response 18: Done
Reviewer 2 Report
Comments and Suggestions for Authors
The trematode Fascioloides magna, originally a parasite of North American deer, has adapted to new host species upon introduction to Europe. This study investigates its infection dynamics in roe deer (Capreolus capreolus)—an aberrant host traditionally believed incapable of developing pseudocysts. Liver samples collected from Bjelovar-Bilogora (BB) and Zagreb (ZG) counties in Croatia over four hunting seasons (2019–2023) were examined for macroscopic liver lesions and parasite developmental stages.
Observations and suggestions:
1.The authors report a mean prevalence of 12.86% in BB and 3.8% in ZG. However, it is unclear whether they refer to point prevalences for each season or year, or cumulative values. Why not seasonal incidence. This needs clarification in both text and figures.
- Figures require substantial improvement for clarity: arrows, numeric annotations, consistent legends, and figure captions that explicitly describe axes and trends are necessary. For instance, Figure 4 introduces the term "linear prevalence" without explanation—this should be defined or rephrased.
- In Figures 7 and 8, it is unclear whether data points refer to calendar years or specific hunting seasons. Additionally, the regression R² values are relatively low and do not justify strong inferential conclusions.
- A key finding states that odds of pseudocyst presence were 2.9 times higher in 2020/21 compared to 2019/20. However, the confidence interval (CI 95%: 0.3163 to 26.8924) is wide and includes 1, and the p-value (p = 0.345) is not statistically significant. This undermines the reliability of the claim. The authors must emphasize that while an increased odds ratio may appear biologically suggestive, the result is not statistically significant, and conclusions regarding host adaptation should remain cautious.
- Furthermore, the sign of regression coefficients (positive or negative) is not consistently reported in the text or figures. This should be clearly indicated, along with exact model parameters.
- The manuscript implies a shift in host-parasite dynamics, with roe deer increasingly developing pseudocysts and possibly surviving longer or repeated infections. While observational trends support this hypothesis, the lack of consistent statistical significance weakens the strength of the conclusion.
This study contributes important observational data on F. magna in roe deer, but the current conclusions regarding adaptation are not fully supported by statistically significant evidence. The paper would benefit from a more cautious interpretation of the results and clearer data presentation.
Comments on the Quality of English Language
The paragraph from lines 201 to 206 needs to be rewritten for greater clarity and correctness. For example: Odds ratio analysis indicated an apparent increase in the likelihood of pseudocyst presence over time; however, none of the results reached statistical significance. Specifically, the odds of pseudocyst detection were 2.9 times higher in the 2020/21 season compared to 2019/20 (OR = 2.9167, 95% CI: 0.3163–26.8924, p = 0.345). In 2021/22, the odds were 2.8 times higher (OR = 2.8000, 95% CI: 0.2809–27.9081, p = 0.380), and in 2022/23, they were 2.7 times higher (OR = 2.7317, 95% CI: 0.3108–24.0095, p = 0.365) compared to 2019/20. Although these values suggest a trend toward increased pseudocyst formation, the wide confidence intervals and p-values above 0.05 indicate that the findings are not statistically significant and should be interpreted with caution.
Author Response
Comment 1: The authors report a mean prevalence of 12.86% in BB and 3.8% in ZG. However, it is unclear whether they refer to point prevalences for each season or year, or cumulative values. Why not seasonal incidence. This needs clarification in both text and figures.
Response 1: We have addressed this issue. It is mean prevalence for all seasons. It is not possible to calculate incidence as we have no data on initial health status of population to be able to calculate new cases of disease. New sentence is now – '' The mean prevalence of positive livers during the whole study period was 12.86% in BB and 3.8% in ZG.''
Comment 2; Figures require substantial improvement for clarity: arrows, numeric annotations, consistent legends, and figure captions that explicitly describe axes and trends are necessary. For instance, Figure 4 introduces the term "linear prevalence" without explanation—this should be defined or rephrased.
Response 2: Thank you for noticing this. Figures are corrected.
Comment 3: In Figures 7 and 8, it is unclear whether data points refer to calendar years or specific hunting seasons. Additionally, the regression R² values are relatively low and do not justify strong inferential conclusions.
Response 3: Thank you for comment. We are aware of that and whole text includes term ''potential''. We have avoided strong inferential conclusions. Figures are corrected.
Comment 4: A key finding states that odds of pseudocyst presence were 2.9 times higher in 2020/21 compared to 2019/20. However, the confidence interval (CI 95%: 0.3163 to 26.8924) is wide and includes 1, and the p-value (p = 0.345) is not statistically significant. This undermines the reliability of the claim. The authors must emphasize that while an increased odds ratio may appear biologically suggestive, the result is not statistically significant, and conclusions regarding host adaptation should remain cautious.
Response 4: Corrected.
Comment 5: Furthermore, the sign of regression coefficients (positive or negative) is not consistently reported in the text or figures. This should be clearly indicated, along with exact model parameters.
Response 5: Corrected.
Comment 6: The manuscript implies a shift in host-parasite dynamics, with roe deer increasingly developing pseudocysts and possibly surviving longer or repeated infections. While observational trends support this hypothesis, the lack of consistent statistical significance weakens the strength of the conclusion.
Response 6: Thank you for comment. We are aware of that, however, biological part of data indicates potential modifications, and we believe that statistics solely is not sufficient to support all trends in biology. Whole text is formulated now in a way to emphasize potential adaptations.
Comment 7: This study contributes important observational data on F. magna in roe deer, but the current conclusions regarding adaptation are not fully supported by statistically significant evidence. The paper would benefit from a more cautious interpretation of the results and clearer data presentation.
Response 7: Done. Explained in previous comment.
Comment 8: Comments on the Quality of English Language
The paragraph from lines 201 to 206 needs to be rewritten for greater clarity and correctness. For example: Odds ratio analysis indicated an apparent increase in the likelihood of pseudocyst presence over time; however, none of the results reached statistical significance. Specifically, the odds of pseudocyst detection were 2.9 times higher in the 2020/21 season compared to 2019/20 (OR = 2.9167, 95% CI: 0.3163–26.8924, p = 0.345). In 2021/22, the odds were 2.8 times higher (OR = 2.8000, 95% CI: 0.2809–27.9081, p = 0.380), and in 2022/23, they were 2.7 times higher (OR = 2.7317, 95% CI: 0.3108–24.0095, p = 0.365) compared to 2019/20.
Response 8: Corrected. English language is revised by authorized office.
Round 2
Reviewer 2 Report
Comments and Suggestions for Authors
I was unable to follow the modifications due to the absence of track changes. The results presented are not supported by statistical significance, and no proven association has been demonstrated. The manuscript may be considered for publication as an observational note on fasciolosis.
Author Response
Reviewer 2
Comment 1: The authors report a mean prevalence of 12.86% in BB and 3.8% in ZG. However, it is unclear whether they refer to point prevalences for each season or year, or cumulative values. Why not seasonal incidence. This needs clarification in both text and figures.
Response 1: We have addressed this issue. It is mean prevalence for all seasons. It is not possible to calculate incidence as we have no data on initial health status of population to be able to calculate new cases of disease. New sentence is now – '' The mean prevalence of positive livers during the whole study period was 12.86% in BB and 3.8% in ZG.''
Comment 2; Figures require substantial improvement for clarity: arrows, numeric annotations, consistent legends, and figure captions that explicitly describe axes and trends are necessary. For instance, Figure 4 introduces the term "linear prevalence" without explanation—this should be defined or rephrased.
Response 2: Thank you for noticing this. Figures are corrected.
Comment 3: In Figures 7 and 8, it is unclear whether data points refer to calendar years or specific hunting seasons. Additionally, the regression R² values are relatively low and do not justify strong inferential conclusions.
Response 3: Thank you for comment. We are aware of that and whole text includes term ''potential''. We have avoided strong inferential conclusions. Figures are corrected.
Comment 4: A key finding states that odds of pseudocyst presence were 2.9 times higher in 2020/21 compared to 2019/20. However, the confidence interval (CI 95%: 0.3163 to 26.8924) is wide and includes 1, and the p-value (p = 0.345) is not statistically significant. This undermines the reliability of the claim. The authors must emphasize that while an increased odds ratio may appear biologically suggestive, the result is not statistically significant, and conclusions regarding host adaptation should remain cautious.
Response 4: Corrected.
Comment 5: Furthermore, the sign of regression coefficients (positive or negative) is not consistently reported in the text or figures. This should be clearly indicated, along with exact model parameters.
Response 5: Corrected.
Comment 6: The manuscript implies a shift in host-parasite dynamics, with roe deer increasingly developing pseudocysts and possibly surviving longer or repeated infections. While observational trends support this hypothesis, the lack of consistent statistical significance weakens the strength of the conclusion.
Response 6: Thank you for comment. We are aware of that, however, biological part of data indicates potential modifications, and we believe that statistics solely is not sufficient to support all trends in biology. Whole text is formulated now in a way to emphasize potential adaptations.
Comment 7: This study contributes important observational data on F. magna in roe deer, but the current conclusions regarding adaptation are not fully supported by statistically significant evidence. The paper would benefit from a more cautious interpretation of the results and clearer data presentation.
Response 7: Done. Explained in previous comment.
Comment 8: Comments on the Quality of English Language
The paragraph from lines 201 to 206 needs to be rewritten for greater clarity and correctness. For example: Odds ratio analysis indicated an apparent increase in the likelihood of pseudocyst presence over time; however, none of the results reached statistical significance. Specifically, the odds of pseudocyst detection were 2.9 times higher in the 2020/21 season compared to 2019/20 (OR = 2.9167, 95% CI: 0.3163–26.8924, p = 0.345). In 2021/22, the odds were 2.8 times higher (OR = 2.8000, 95% CI: 0.2809–27.9081, p = 0.380), and in 2022/23, they were 2.7 times higher (OR = 2.7317, 95% CI: 0.3108–24.0095, p = 0.365) compared to 2019/20.
Response 8: Corrected. English language is revised by authorized office.
Round 3
Reviewer 2 Report
Comments and Suggestions for Authors
The manuscript provides a valuable and timely investigation into the evolving host-parasite relationship between Fascioloides magna and roe deer in two distinct regions of Croatia.
Suggestions:
1. The conclusion that roe deer may be developing the ability to survive multiple infections with F. magna, as indicated by the formation of pseudocysts, is a significant insight. But it is just a sugesstions not a proved fact (not in terms of the scientificaly reductionism) . The manuscript would benefit from a short discussion on possible mechanisms behind this shift—whether it may involve immunological adaptation, parasite attenuation, or other ecological factors.
I recommend ensuring uniformity in graph design, including consistent axis labeling (font size, style), color schemes, and legend placement throughout all figures. All graphs should ideally follow the same shape and format (e.g., bar charts vs. line graphs) where applicable, to enhance visual coherence and ease of comparison across years or regions.
Ensure consistent use of terminology (e.g., “positive livers” vs. “infected livers”).
The odds ratio (2.8) is important—please include confidence intervals and p-values to provide better statistical context for readers. It have to be included....otherwise the readers will consider that this is a proved odds ration ... and you know it si not significant.
Author Response
- The conclusion that roe deer may be developing the ability to survive multiple infections with F. magna, as indicated by the formation of pseudocysts, is a significant insight. But it is just a sugesstions not a proved fact (not in terms of the scientificaly reductionism) . The manuscript would benefit from a short discussion on possible mechanisms behind this shift—whether it may involve immunological adaptation, parasite attenuation, or other ecological factors.
At this moment it is very difficult to discuss this topic in detail without scientific confirmation. A part of explanation was already included from an article published by Csivincsik et al., and one new sentence was added about host's immune responses.
- I recommend ensuring uniformity in graph design, including consistent axis labeling (font size, style), color schemes, and legend placement throughout all figures. All graphs should ideally follow the same shape and format (e.g., bar charts vs. line graphs) where applicable, to enhance visual coherence and ease of comparison across years or regions.
Done.
3, Ensure consistent use of terminology (e.g., “positive livers” vs. “infected livers”).
Done. Changed to infected.
- The odds ratio (2.8) is important—please include confidence intervals and p-values to provide better statistical context for readers. It have to be included....otherwise the readers will consider that this is a proved odds ration ... and you know it si not significant.
We have added range for odds that was obtained during the study, with CI and p value.